# Characterization and Cell Viability of Probiotic/Prebiotics Film Based on Duck Feet Gelatin: A Novel Poultry Gelatin as a Suitable Matrix for Probiotics

**DOI:** 10.3390/foods10081761

**Published:** 2021-07-30

**Authors:** Ahmadreza Abedinia, Faezeh Alimohammadi, Farangis Teymori, Najibeh Razgardani, Mohammad Reza Saeidi Asl, Fazilah Ariffin, Abdorreza Mohammadi Nafchi, Nurul Huda, Jumardi Roslan

**Affiliations:** 1Food Biopolymer Research Group, Food Science and Technology Department, Damghan Branch, Islamic Azad University, Damghan 36711, Iran; raabediniar@gmail.com (A.A.); farangisteymoori66@gmail.com (F.T.); 2Young Researchers and Elite Club, Shahr-e-Qods Branch, Islamic Azad University, Tehran 37541, Iran; alimohammadi.faezeh@yahoo.com; 3Department of Food Science and Technology, Garmsar Branch, Islamic Azad University, Garmsar 35811, Iran; najibehrazgordani@gmail.com; 4Department of Food Science and Technology, Sabzevar Branch, Islamic Azad University, Sabzevar 96131, Iran; dr.saeidi@iaus.ac.ir; 5Food Biopolymer Research Group, Food Technology Division, School of Industrial Technology, Universiti Sains Malaysia, Minden 11800, Penang, Malaysia; 6Faculty of Food Science and Nutrition, Universiti Malaysia Sabah, Kota Kinabalu 88400, Sabah, Malaysia; jumardi@ums.edu.my; 7Department of Food Science and Technology, Faculty of Agriculture, Universitas Sebelas Maret, Surakarta 57126, Central Java, Indonesia

**Keywords:** duck feet gelatin, synbiotic edible film, probiotic viability

## Abstract

The probiotic viability, physicochemical, mechanical, barrier, and microstructure properties of synbiotic edible films (SEFs) based on duck feet gelatin (DFG) were evaluated. Four synbiotic systems were obtained by mixing four types of prebiotics, namely, dextrin, polydextrose, gum Arabic, and sago starch, with DFG to immobilize of probiotic (*Lactobacillus casei* ATCC). The ability of DFG to create a suitable matrix to increase probiotic viability was compared with those of other commercial gelatins in a preliminary evaluation. The DFG showed proper probiotic viability compared with other gelatins. The addition of prebiotics reduced the transparency of SEFs and increased color differentiation, uniformity, and complete coverage of probiotic cells. The estimated shelf-life of surviving bacteria in the SEFs stored at 4 and 25 °C showed that gum arabic showed the best performance and enhanced the viability of *L. casei* by 42% and 45%, respectively. Dextrin, polydextrose, and sago starch enhanced the viability of *L. casei* at 4 and 25 °C by 26% and 35%, 26% and 5%, and 20% and 5%, respectively. The prebiotics improved the physicochemical, mechanical, and barrier properties of all SEFs, except polydextrose film. The viability of *L. casei* can be increased with the proper selection of gelatin and prebiotics.

## 1. Introduction

The probiotics global market in 2014 was valued at 62.6 billion dollars; in 2020, it was valued at 96.0 billion dollars [1]. Probiotics are live microorganisms that confer health benefits on the host through the maintenance of normal intestinal microflora; suitable probiotic organisms are administered at a sufficient number, i.e., >6–7 log colony-forming unit (CFU)/g. However, some studies have reported limitations in terms of the probiotics dose. The appropriate dose for the effective use of probiotics is less than (10^9^ CFU/day). Therefore, the validity of studies that do not consider the minimum consensus of appropriate probiotic dosages in their results is questioned [2]. Some of the probiotic benefits include the following: enhancement of the immune system, which in turn affects the severity and prevention of and susceptibility to COVID-19 [3]; protection against gastrointestinal pathogens [4]; reduction of serum cholesterol level and blood pressure [5]; anti-carcinogenic activity [6]; and the improvement of absorption and utilization of nutrients from food [7,8]. Probiotics have many therapeutic applications, as reported by Dhillon and Singh [9]. Generally, Lactobacilli, Bifidobacteria, and *Saccharomyces* are generally recognized as safe (GRAS) by the WHO [10]. 

One of the biggest problems that leads to failure is the viability of the required number of probiotic bacteria in the path of production and in the digestive tract pathway [11]. The content of probiotic bacteria in food right before ingestion is suggested to be in the range of 10^8^–10^9^ CFU /g to ensure the adequate therapeutic minimum of 10^6^–10^7^ CFU/g colon in the final product [12]. 

Various methods have been developed to increase the survival of probiotics, among which microencapsulation is the most useful. The combination of microencapsulation with active edible packaging represents a promising strategy for the adequate protection and delivery of probiotic species [1,13]. The production of edible films and coatings as stabilizing matrices of probiotics allows them to reach the intestines in sufficient quantities [14]. 

Various biopolymers have been used as carrier matrices of probiotics, such as the following: kefiran [15], low esterified amidated pectin, kappa-carrageenan/locust bean gum and gelatin [16], carboxymethyl cellulose (CMC) [17], poly(vinyl alcohol), hydroxypropyl methylcellulose and corn starch [18], CMC and hydroxyethyl cellulose [19,20], gum Arabic (GA), maltodextrin and whey protein concentrate (WPC) [21], cassava starch (CS)/CMC [22,23], CMC nanofibers [24], chitosan-coated agar-gelatin gel particles [25], and Konjac glucomannan [26].

Gelatin is a very suitable vehicle for encapsulating bioactive compounds, such as probiotics living cells [27] and microencapsulating agents [28]. Poultry gelatins, such as duck feet gelatin (DFG) and chicken skin gelatin (CSG), have excellent film-forming properties [29,30]. One of the primary functions of food packaging is to reduce moisture transition between products and the surrounding atmosphere. Low water-vapor permeability (WVP) widens the composite packaging film utilization, especially in a high-moisture environment [31]. The WVP value of CSG film was low at 1.36 × 10 **^−^**^4^ g m^−1^ s^−1^ Pa^−1^. That of DFG was 5 × 10 **^−^**^11^ g m^−1^ s^−1^ Pa^−1^ [32], whereas that of bovine gelatin (BG) film was 2.5 × 10^−10^ g m^−1^ s^−1^ Pa^−1^ [33]. Poultry gelatin, when extracted optimally, contains more than twice as much collagen as commercial gelatin, which makes poultry gelatin more economically and nutritionally cost effective [34].

Prebiotic is a nondigestible ingredient that beneficially affects the host by selectively stimulating the growth and/or activity of one or a limited number of bacteria in the colon without improving the well-being and health of unwanted bacteria. Some biopolymers, including gum arabic (GA), polydextrose (Poly), dextrin (Dex), and starch, have a prebiotic effect on food systems. The synergistic combinations of probiotics and prebiotics are called synbiotics [26]. 

This study aimed to develop and characterize DFG-based bioactive films containing *L. casei* ATCC 393 cells incorporated with four prebiotics, namely, sago starch (SS), GA, Poly, and Dex for the first time. This bioactive film can be used as an edible coating or film of ready-to-eat meat products, processed cheeses, and dried fruits.

## 2. Materials and Methods

### 2.1. Materials, Bacterial Strains, and Culture Conditions

BG (G9382; ~225 g Bloom) and prebiotic fibers, such as dextrin from maize starch (CAS No. 9050-36-6) and polydextrose (Promitor), were purchased from Sigma-Aldrich and Tate and Lyle GmbH, (Darmstadt, Germany), respectively. The DFG was extracted by acetic acid according to the method described by Abedinia, Ariffin, Huda, and Nafchi [29] at the School of Industrial Technology, Universiti Sains Malaysia. Fish gelatin (FG) was extracted from tilapia skin gelatin according to the method of Tan et al. [35]. Gum arabic (molecular weight: ~250,000) and sago starch (10% moisture; a mean diameter of 37.59 μm of granules) were purchased from SIM Supplies Company (Penang, Malaysia).

Lyophilized commercial cultures of *L. casei* ATCC 393 used for immobilization were obtained from DSMZ, Germany. Preparation of stock culture was performed as described by Sidira et al. [36] The probiotic was grown on MRS broth (Merck, Darmstadt, Germany) at 37 °C for 72 h. The cell culture broth in the stationary bacterial growth stage was aseptically transferred to sterile 50 mL plastic centrifuge tubes (Sarstedt Ltd., Leicester, UK) and centrifuged at 3000× *g* for 5 min. The supernatant liquid was discarded, and the harvested bacterial cells were washed twice with phosphate-buffered saline (PBS) (Dulbecco A, Oxoid Ltd., Basingstoke, UK).

### 2.2. Preparation of the Probiotic and Synbiotic Films

The probiotic gelatin films (Pro-DFG, Pro-BG, and Pro-FG) were prepared. The ability of DFG to create a suitable and protective matrix to increase probiotic viability was compared with that of other commercial gelatins. Gelatins (DFG, BG, and FG) at 4 g each were weighted in a 200 mL Duran Schott bottle, and 1 g glycerol (equivalent of 25% of biopolymer; ≥99%, Fischer Scientific, Loughborough, UK) was weighed in a beaker. Glycerol was dissolved in 50 mL of distilled water then mixed with the gelatins. The mixture was heated for 30 min at 50 °C for hydration, and film-forming solutions (FFSs) were obtained.

To develop a DFG bioactive film, the synbiotic edible films (SEFs) (Syn-DFG/Dex, Syn-DFG/Poly, Syn-DFG/GA, and Syn-DFG/SS) were prepared. The prebiotics were added to 50 mL of distilled water at 50 °C at 2 g. In a separate beaker, 4 g of DFG was added to create five individual biopolymer solutions. Glycerol as a plasticizer was added to all solutions at 25% of biopolymer total solid material. DFG solution was heated for 30 min at 50 °C for hydration and mixed at 1:1 with the prebiotic solutions to make FFSs.

The FFSs were neutralized by pH adjustment at 7.0 with sodium hydroxide 0.1 M. Pathogenic bacteria were killed, and clear FFSs were obtained in both probiotic films and SEFs. Bottles were stirred and heated on a Heidolph magnetic stirrer for 15 min at 80 °C. The heated FFSs were cooled at 39 °C and retained isothermally to avoid gelatin setting point until inoculation with probiotics. Three pellets of *L. casei* (corresponding to ca. 10 log CFU/g of film-forming solution-dry basis) were added to each FFSs (100 mL) and degassed using a vacuum pump at 40 °C for 10 min. Then, 45 g of FFS was aseptically transferred using a serologic pipette and cast to Perspex plates (20 × 20 cm^2^) fitted with a square frame around the edge to yield a 16 × 16 cm^2^ film-forming area. The cast solutions were air-dried at 37 °C for 15 h in a ventilated incubator to obtain films that could easily be peeled off. After drying, the films were peeled and conditioned at room temperature (25 ± 1 °C) or chilled at 4 ± 1 °C under controlled relative humidity conditions (54% RH) in desiccators containing saturated magnesium nitrate solution [37].

### 2.3. Microbiological Analysis

To assess DFG’s effect on the viability of probiotics and to compare this effect with those of BG (a mammalian source) and FG (a mammalian alternative gelatin source), a method described by Soukoulis et al. [38] was used with minor modification. The number of viable *L. casei* in the PGFs was evaluated. The number of live *L. casei* in the SEFs was measured every 7 days of storage at 4 and 25 °C in FFSs and compared with the initial amount in films.

To determine the initial number of live probiotics in FFSs, 1 mL of film sample from each of FFS was suspended and vortexed in sterile PBS for half a minute to ensure thorough mixing. For PGFs, 1 g of each film sample was moved to 9 mL of sterile PBS. Then, individual pieces were left for hydration and then dissolved under constant agitation in an incubator at 37 °C for 1 h. To determine the effect of edible films without probiotics, the edible films were completely dissolved until no residual insoluble material was recognizable. By using PBS, the obtained solutions were subjected to serial dilutions. Every dilution was pour-plated on MRS agar (Oxoid Ltd., Basingstoke, UK), and the plates were kept for 72 h at 37 °C under anaerobic situations. Bacteria enumeration on agar plates was accomplished in triplicates by colony counting. All viable bacteria counts were expressed as log colony-forming units per gram (log CFU/g, CFU/g = CFU/plate dilution factor). The survival rate of the bacteria throughout the film-forming solution-drying procedure was obtained according to the following equation:(1)$$\%\mathrm{Viability}=\frac{N}{N_{0}}\times 100$$
where *N*_0_ and *N* represent the numbers of viable bacteria before and after the implemented drying process, respectively [16].

*L. casei* inactivation upon storage data was expressed as the value of the relative viability fraction *N*/*N*_0_. The viability data were fitted to a first-order reaction kinetics model, as described by the following equation:(2)NtN0=1-kTt Equation 
where: 

*N*_0_ represents the initial number of viable bacteria

*N_t_* is the number of viable bacteria after a specific storage time (in CFU/g)

*t* is the storage time (in a day)

*k_T_* is the inactivation rate constant at *T* temperature (day^−1^).

### 2.4. Physicochemical Properties

Analysis of the SEFs was performed after film conditioning in a desiccator containing saturated salt at a relative humidity of 54% and temperature of 25 °C for 3 days.

#### 2.4.1. Thickness and Moisture Content (MC) 

Film thickness was measured by using a hand-held micrometer with an accuracy of 0.001 mm (thickness gauge; Ozaki MFG Co., Tokyo, Japan). Five measurements were taken from random samples and used to determine the average thickness.

The *MC* was measured using the method described by Li, Ma, Ji, Sameen, Ahmed, Qin, Dai, Li, and Liu [22] with slight modification and calculated according to the following equation:(3)$$\mathrm{MC}\;(\%)=\frac{m_{1\;}{-\;m}_{2}}{m_{1}}\times 100$$
where *m*_1_ and *m*_2_ are the initial and final weights of the SEFs, respectively.

#### 2.4.2. Water Solubility (WS) and Swelling Ratio (SR)

Film specimens (2 cm × 2 cm) desiccated by P_2_ O_5_ for 7 days were used for both measurements. The *WS* was determined based on the method described by Singh, Magalhães, Alves, Antunes, Miguel, Lindman, and Medronho [19] with slight modification. The samples were weighed (*W*_0_) and immersed in 80 mL of deionized water (18 MΩ) at 35 ± 1 °C under 50 rpm agitation for 1 h. Unresolved SEFs were dried to a constant weight (*W_1_*) at 60 °C. The *WS* was expressed in the following the equation:(4)$$\mathrm{WS}\;\left(\%\right)=\frac{w_{0\;}{-\;w}_{1}}{w_{0}}\times 100$$

The *SR* was measured according to the method of Abedinia, Ariffin, Huda, and Nafchi [32] with slight modification. Dried specimens were weighed as initial dry matter (*W*_1_), immersed in 30 mL of distilled water, and kept at 25 °C for 2 h. The surplus water on the surface was removed by using a filter paper. Afterward, the samples were weighed immediately (*W*_2_). The *SR* was calculated according to the following equation:(5)$$\mathrm{SR}=\left(\%\right)=\frac{w_{2\;}{-\;w}_{1}}{w_{1}}\times 100$$

Experiments were performed in five replicates, and the average was reported as the *SR* value. 

#### 2.4.3. Color and Opacity Properties 

Applying a Minolta colorimeter (Chroma Meter CR-400/410, Konica Minolta, Japan), color parameters were determined at three different areas of the sample’s surface after calibration by using a white plate (*L^*^* = 93.58, *a^*^* = −0.88, and *b^*^* = 0.46). The color was recorded by using *CIE-L^*^ a^*^ b^*^* uniform color space (CIE-Lab), where *L^*^* demonstrates lightness, *a^*^* shows hue on a green (–) to red (+) axis, and *b^*^* exposes the hue on a blue (–) to yellow (+) axis. The total color differentiation *ΔE^*^* between the control sample (film without prebiotics) and SEFs was counted using the following equation:(6)$${\Delta E}^{*}=\sqrt{{(\Delta L}^{*}{)\;}^{2}{+(\Delta a}^{*\;}{)\;}^{2}{+(\Delta b}^{*}{)\;}^{2}}$$

Opacity measurements were obtained based on the method devised by Soukoulis, Behboudi-Jobbehdar, Macnaughtan, Parmenter, and Fisk [16] using a UV-VIS spectrophotometer (Jenway Ltd., Stone, UK). The absorbance at 550 nm (*A*_550_) was calculated, and film opacity was obtained based on the following formula:(7)$$\mathrm{Opacity}\;\mathrm{value}=\frac{A_{550}}{\mathrm{Thickness}}$$
where the thickness is expressed in mm.

#### 2.4.4. Water-Vapor Permeability (WVP)

The gravimetrically ASTM E96-16 standard method was applied to determine *WVP* at 22 ± 1 °C. Weight loss versus time was plotted to obtain the slope (r^2^ ≥ 0.99). The *WVP* of the film was calculated by Equation (8).
(8)$${\mathrm{WVP}\;(g\;m}^{-1}{\;s}^{-1}{\;\mathrm{Pa}}^{-1}{)\;}_{\;}=\frac{\mathrm{WVTR}\times L}{\Delta P}$$
where:

*WVTR* is the rate of water-vapor transmission (g/m^2^ s^1^) through film,

*L* is the average film thickness (m),

Δ*P* is the partial water-vapor pressure difference (Pa) between the two sides of the film. Three measurements were made for each film, and the mean value was reported. 

#### 2.4.5. Prevention against Oxidation (PAO)

According to the method described by Akman et al. [39], the PAO of the SEFs was measured indirectly with slight modification. Samples were conditioned at 25 °C, 55% RH for 72 h. Free antioxidant (10 mL), namely, fresh and refined sunflower oil, was poured into the cup and completely covered by each film and stored for 45 days inside an incubator at 25 ± 1 °C and 55% RH. The oil peroxide values (PVs) were determined by using the method described by AOCS [40] and reported as the PAO index. 

### 2.5. Mechanical Properties of SEFs

The mechanical properties were measured according to ASTM D882-16 with minor modification by applying a texture analyzer (TA.XT2i Texture Analyzer; Stable Micro Systems, Godalming, Surrey) equipped with a 30 kg load cell. The tests included the evaluation of tensile strength (TS), Young’s modulus (YM), and elongation at break (EB).

### 2.6. Scanning Electron Microscopy (SEM) of SEFs

The microstructure of films was studied by SEM (Everhart–Thornley detector (FE-SEM-ETD) with QUANTA FEG 650 2012 SEM system, FEI, Hillsboro, OR, USA). To facilitate SEM measurements, the samples with dimensions of 1 cm × 1 cm from each film were coated with chromium, and their surface morphology was determined with an accelerating voltage of 5.0 kV.

### 2.7. Statistical Analysis

Statistical analyses were performed by statistical software package SPSS Version 24 (IBM Corp., Armonk, NY, USA). To determine the significant effects of prebiotics on physicochemical and mechanical properties of SEFs, the obtained experimental data were subjected to one-way analysis of variance (ANOVA) followed by an analysis of the comparison of means by using Duncan’s post hoc method. The significance level was *p* < 0.05.

## 3. Results and Discussions

### 3.1. Effect of Gelatin Origin on the Storage Stability of L. casei 

The purpose of this evaluation was to compare the ability of various gelatins (DFG, BG, and FG) in film form to enhance the survival of *L*. *casei* at 25 °C. The inactivation curves of *L. casei* immobilized in three different types of gelatin are shown in Figure 1 and Table 1. As the curves show, DFG has the potential to keep *L. casei* alive. 

Dong et al. [41] declared that one of the reasons for the inactivation of the live probiotics encapsulated in the biopolymer matrix is the matrix structure; physical structure can lead to the molecular mobility of the solvent throughout the matrix. Soukoulis et al. [42] evaluated the *L. rhamnosus* GG stability in binary starch–protein edible films. The gelatin showed the highest protection against osmotic and heat stress-induced injuries during drying, particularly in rice-based films. Proteins can increase probiotic survival by supplying micronutrients and scavenging free radicals (such as amino acids and peptides) that are essential for improving the weak growth of proteolytic probiotic bacteria [43,44]. The imino acid contents (proline + hydroxyproline) of gelatin stabilize the structure by forming hydrogen bonds. The hydroxyproline content of DFG is higher than that of mammalian and chicken gelatins [27]. Table 1 shows that the shelf-life of bacteria in DFG gelatin matrix was higher than in the other two gelatins, but the difference was not statistically significant. The results of this experiment indicated that DFG had good potential for immobilizing bacteria and was a good option as a film synbiotic.

### 3.2. Color Properties and Morphology

In the present study, homogeneous and flexible films were obtained after drying at 25 °C and 55% RH. Results of measurement of the appearance properties provide information about the films’ intended application but not the chemical reaction. Film transparency is the key to solving the problem of good film acceptance by end-users when the film is used to cover or improve the food surface. Table 2 shows the optical and color properties of edible bioactive films. It shows that the addition of prebiotic fiber was associated with a detectable decrease (*p* < 0.05) of the transparency (or increase in opacity) of the edible films compared with those that only contain DFG. This observation was in accordance with the results of Soukoulis et al. [45] and Wu et al. [46].

Significant differences in the luminosity (*L^*^*) of the SEFs (*p* < 0.05) were observed. The control sample and Syn-DFG/Dex film exhibited the highest and lowest luminosity, respectively, and no significant differences were observed among gum GA, Poly, and Syn films. When prebiotics were added, significant differences in *a^*^* and *b^*^* were observed in all SEFs. Syn-DFG/Poly exhibited the highest scores for green and yellow color components, whereas Syn-DFG/Dex and control showed the highest values for red and blue hue color components (*a^*^* and *b^*^*), respectively.

*ΔE^*^* results indicated that in all cases, *ΔE^*^* values were higher than 3, which was assumed to be the threshold of human-perceivable color difference [26]. Syn-DFG/SS had the lowest color divergence from the control, whereas Syn-DFG/Poly had the highest. Thus, according to the results, opacity and color differences with significant probability could be attributed to prebiotic fiber presence and type. The results are shown in Figure 2. 

The SEM microscopic analysis of the SEFs (Figure 3) indicated that the addition of prebiotic fibers in the DFG film leads to noticeable changes in the microstructure of the SEFs. SEM micrographs show that blending the prebiotic fiber with gelatin before film formation leads to a more uniform and compact structure with no noticeable micropores or interspaces. Prebiotics act as fillers of the interspaces of entangled gelatin network. *L. casei* cells on the probiotic edible film surfaces were not identified (data not shown). The extensive coverage and properties of bacterial cells in SEFs were better than those in gelatin only. No remarkable differences between the structural conformations of the films comprising Dex, GA, Poly, and SS were detected, as mentioned by previous researchers [16,47].

The Syn-DFG/SS and Syn-DFG/GA showed a more nonporous, smooth, and uniform structure than the Poly and Dex films. However, in all cases, prebiotics application resulted in good miscibility and compatibility with gelatin, possibly through hydrogen-bond interactions, because no phase aggregation or separation phenomena were observed. Further studies are needed to completely characterize the phase compatibility within the biopolymers, as this topic was not among the initial objectives of this research.

### 3.3. Effect of Prebiotics on the Storage Stability of L. casei 

In Figure 4, the inactivation curves of *L. casei* immobilized in SEFs are presented. As expected, the inactivation rates (Table 3) of *L. casei* were considerably higher (*p* < 0.05) in the systems preserved at room temperature. Keeping the SEFs at 25 °C in the presence of GA and Dex in the plasticized matrices improved the storage stability of *L. casei*. Moreover, SS and Poly had no significant effect on the improvement of *L. casei*. The synbiotic film containing GA had the highest estimated shelf-life values at both storage temperatures according to its ability to sustain *L. casei*, followed by Dex, Poly, and SS. The increase in storage temperature induced an almost four-fold acceleration of the inactivation rate of *L. casei*. Calame et al. [48] demonstrated that the numbers of Bifidobacteria and Lactobacilli 4 weeks after consumption were significantly higher for GA. One of the most promising applications related to GA is that it is relatively impossible for various enzymes in the small intestine to access it. Thus, GA is a prebiotic that can be categorized as nondigestible food. GAs are substances that contribute to the health of the host by selectively stimulating the growth and/or activity of one bacteria or a limited number of bacteria in the large intestines without stimulating unwanted bacteria.

The estimated shelf-life of the SEFs (in terms of *L. casei* survival) was in the range of 78–111 and 20–29 days for the systems preserved under 4 and 25 °C conditions, respectively (Table 3). External factors, such as oxygen, water activity, and temperature, are known to negatively affect the viability of living probiotic cells. In the case of intermediate moisture systems that contain edible films, the presence of high amounts of solutes together with the rubbery physical state (solutes improved molecular mobility) facilitates the occurrence of chemical and enzymatic reactions that destroy essential cellular structures, e.g., phospholipid membrane bilayers [49]. Therefore, using low *MC* matrices with low permeability to gases comprising free radical scavenging agents (to control lipid oxidation of cellular membranes) is a proficient strategy for increasing probiotic viability in food systems [50]. Generally, the prebiotic film constancy at room temperature is comparable to that of anhydrobiotics, e.g., spray-dried powders, under the same relative humidity conditions [51].

A complete mechanistic understanding of probiotics stability is not available in biopolymer matrices during storage. However, factors such as steric hindrance of solutes, the translational matrix diffusion of oxygen, the presence of nutrients and free radical scavenging agents, and the interaction via hydrogen bonding with the polar head groups of membranes phospholipids can possibly contribute to the stability of probiotics in prebiotic films [52,53]. 

### 3.4. Effect of Prebiotics on Physicochemical Properties of SEFs

Table 4 shows the results for thickness, MC (moisture content), SR (percentage of water gained [g]/total solids [g]), and WS (percentage of soluble solids [g]/total solids [g]) of SEFs. With the addition of prebiotics to DFG, a considerable (*p* < 0.05) increase was found in film thickness. Except for GA, the type of prebiotics did not considerably affect the thickness of the edible films. However, a difference was observed between the DFG only films and the others. Our findings are in agreement with those of Soukoulis, Singh, Macnaughtan, Parmenter and Fisk [42], Galus et al. [54], and Fakhouri et al. [55]. 

In terms of MC, Syn-DFG/SS showed a significant difference (10.95%) from the others. Water-vapor mass transfer through the film is the driving force of moisture distribution from the film. Prebiotic type did not exert significant influence on the *MC* of samples. Generally, the concentration, structuring ability of the biopolymers, and water-holding capacity, combined with the amount and kind of plasticizing agents, are the main parameters influencing equilibrium moisture levels in edible films [16]. 

The MC of probiotic products is another factor influencing the shelf-life stability of live bacteria. Oxygen and moisture adversely affect bacterial survival. The amount of remaining water affects bacterial survival during drying and storage [56]. Braber, Vergara, Rossi, Aminahuel, Mauri, Cavaglieri, and Montenegro [50] suggested that molecular mobility of the matrix composition and moisture uptake features are better factors to consider for improving probiotic viability during storage. 

The WS values found for DFG, Dex, and SS films (19.37, 18.08, and 18.5% g soluble solids/g total solids) were considerably lower than the values found for the GA and Poly films (29.49 and 44.07% g soluble solids/g total solids). This finding possibly reveals that Dex, SS, and control films are more resistant to dissolution in water than the others. The *WS* is one of the most significant properties investigated in food and pharmaceutical applications. Unlike water permeability, WS is characterized by chemical structure and defines the material tolerance or resistance to water. Therefore, stability in water was indicated. Lee et al. [57] stated that films could be used as covering materials to constrain exudation in frozen foods. The higher the *WS* results are, the more insufficient the stability of such films is. The *SR* defines the amount of water absorbed by films. It reflects significant property features of carbohydrate and protein films, as these biopolymers primarily swell when suspended in water and show consequential structural changes. The SR has been used to represent the amount of crosslinking and is used in protein films based on collagen. In the current research, no significant differences (*p* < 0.05) in SR were observed among the Poly and the control films. The *SR* values (444.34–675.55%) were obtained. Blended GA and SS had significant (*p* < 0.05) differences with the control, and Dex films showed the lowest *SR*.

### 3.5. Effect of Prebiotics on WVP and PAO of SEFs

The films’ *WVP* was in the range of 2.74–3.37 (×10^−11^.g.m^−1.^ Pa^−1^.s^−1^), thereby showing that the differences depended on the type of prebiotic added. As shown in Figure 5, the addition of all prebiotics except SS was associated with a significant reduction in *WVP* (*p* < 0.05). The highest amount of *WVP* was observed in SS, and the lowest was found in Dex films. The nature of each prebiotic can be the main cause of this discrepancy. The concentrations of prebiotics, gelatin, and glycerol were the same in the films. Thus, the main reason for the difference in the *WVP* was the nature and type of prebiotics. Ebrahimi, Mohammadi, Rouhi, Mortazavian, Shojaee-Aliabadi, and Koushki [17] stated that external factors, such as *WVP*, under normal storage conditions affect immobilized probiotic survival of bacteria in the film structure. Other factors, such as hydrophobicity ratio, crystalline and amorphous ratio, film thickness, and integrity, are also mentioned as practical factors. Adding *L. casei* to DFG film reduced the *WVP* from 5 in our previous study [32] to 3.37 (×10^−11^.g.m^−1.^ Pa^−1^.s^−1^). In general, the difference in *WVP* is directly related to the affinity of film for water; in particular, the weak inhibitory properties of gelatin films affect their tendency to have high water content (Table 4). The lowest amount of *WVP* was observed with the lowest amount of *MC* (Syn-DFG/Dex), thereby showing that dextrin’s ability to reduce the intermolecular space creates hydrogen bonds with gelatin and ultimately reduces water mobility and *WVP*.

The PAO was expressed through SEFs based on the PV. The PV in Figure 6 shows that the PV films varied from 5.01 to 15.41 meq/kg oil compared with the control sample (14.46 meq/kg oil). All combinations of prebiotics, except sago starch, caused a significant reduction (*p* < 0.05) in PV. However, the amount of PV in the SS film was higher than the control, but no statistically significant difference was found (*p* > 0.05). The presence of oxygen around probiotic bacteria can cause toxic metabolites in cells, leading to oxidative stress, damage, and cell death. Therefore, oxygen and its derivatives are effective factors that determine the survival of probiotics [58]. Thus, films that showed the least PAO provided better protective effect for *L. casei*.

### 3.6. Mechanical Properties of SEFs

Mechanical characterization of synbiotic films (Table 5) showed that Dex film had the highest TS (18.36 MPa), followed by GA (18.2 MPa), but these results did not have a significant difference of 5% with the control sample (16.81 MPa) and had a significant difference (*p* < 0.05) with the SS and Poly films. The control sample did not show any difference with the SS film. The films showed EB values ranging between 25.93% for Syn-DFG/Dex and 94.64% for Syn-DFG/Poly films. No statistically significant difference was found between GA and SS films for elongation values, but these films showed significant differences with the control.

Table 5 shows that although the addition of dextrin did not significantly differ with the control in terms of EB, such an addition made the bioactive film more frangible. In the presence of other prebiotics, significant differences were found. The highest EB of 94.64% was observed in Poly films. 

Espitia, Batista, Azeredo, and Otoni [1] confirmed that the addition of prebiotics affected the physical properties of the edible films, which showed impaired tensile strength and boosted extensibility compared with prebiotic-free films (control). In general, except for Syn-DFG/Poly, all of the prebiotics exhibited an acceptable mechanical characterization.

## 4. Conclusions

Films made from DFG could replace gelatin derived from mammals in the generation of bioactive films. The low moisture content of DFG films suggested that they can be a carrier of bioactive compounds, especially probiotics. This study showed that the microstructure of the obtained film from DFG could prevent water mobility, oxygen, and light from reaching the bacteria and can be a more suitable matrix for the probiotic carrier compared with bovine or fish gelatin films. The results of adding prebiotics showed that the gum arabic coatings and fillings were better than others and kept the bacteria alive for longer in two periods of time and at different temperatures. 

Supplementary studies and suggestions are as follows: Several probiotic bacteria can be simultaneously used under the conditions of this study to evaluate the films’ ability to keep bacteria alive.The application of bioactive Syn-DFG/GA film to ready-to-eat or processed-meat packaging needs to be investigated.

## Figures and Tables

**Figure 1 foods-10-01761-f001:**
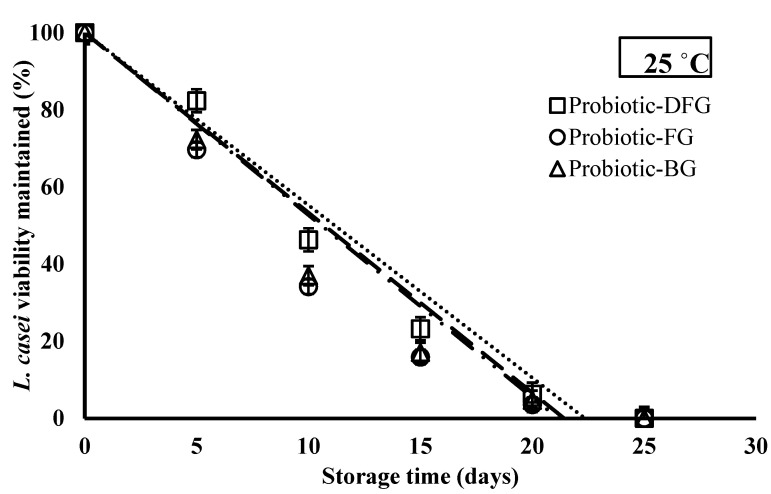
Inactivation curves of *L. casei* during storage at room temperature (25 °C) in different gelatin matrices for 25 days. DFG: Duck feet gelatin, FG: Fish gelatin, BG: Bovine gelatin.

**Figure 2 foods-10-01761-f002:**
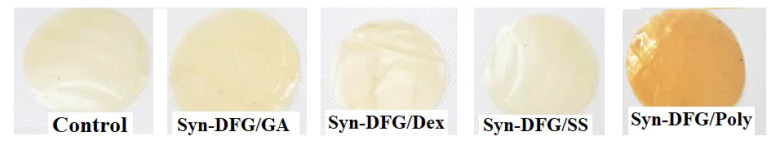
Appearance of probiotic and synbiotic films. Duck feet gelatin, Dex: Dextrin, GA: gum Arabic, Poly: polydextrose, SS: sago starch.

**Figure 3 foods-10-01761-f003:**
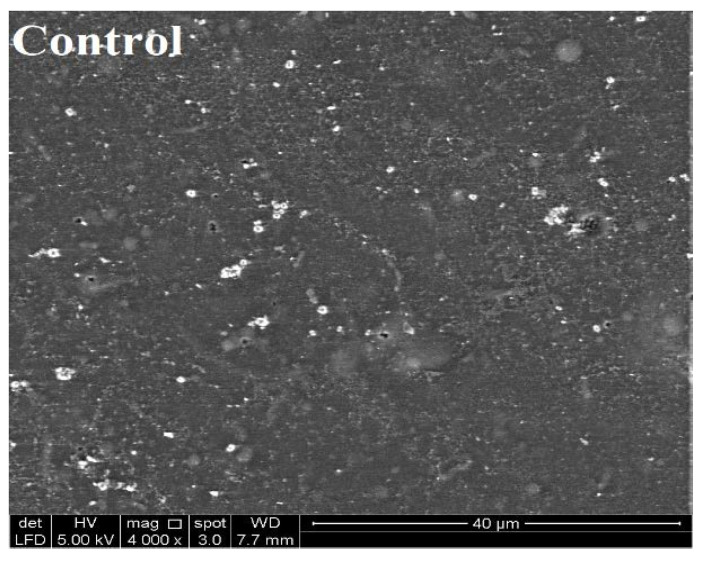
Surface SEM visualization of the probiotic and synbiotic edible films containing *L. casei*. (**a**) Syn-DFG/SS, (**b**) Syn-DFG/GA, (**c**) Syn-DFG/Dex, and (**d**) Syn-DFG/Poly.

**Figure 4 foods-10-01761-f004:**
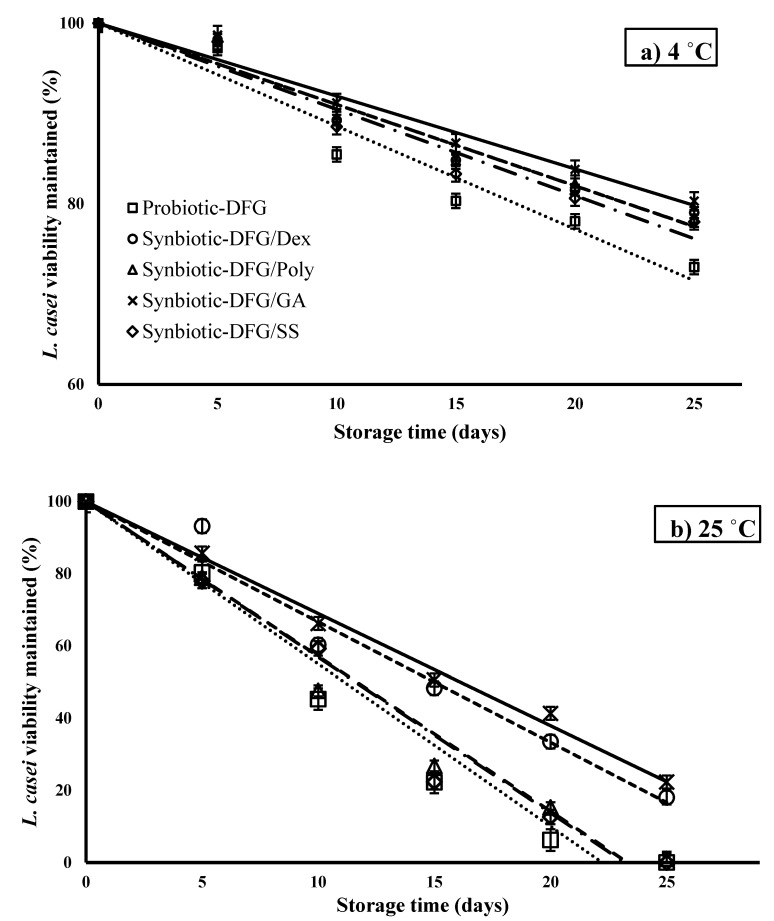
Inactivation curves of *L. casei* trapped in synbiotic edible films during storage: (**a**) under chilled conditions (4 °C) and (**b**) under room temperature conditions (25 °C) for 25 days.

**Figure 5 foods-10-01761-f005:**
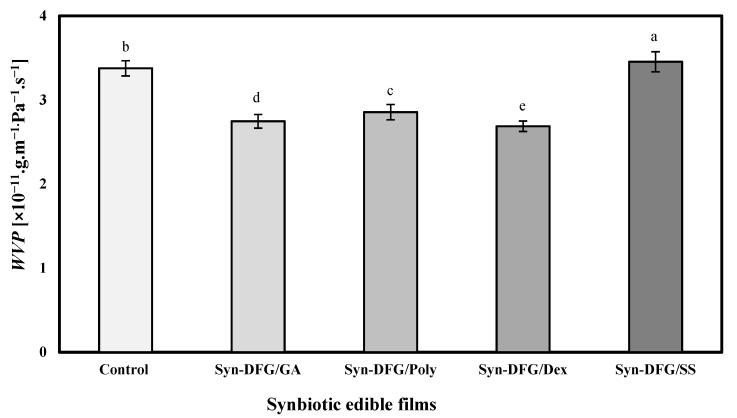
Water-vapor permeability of synbiotic edible films. a–e: Different letters on each column indicate significantly different values (*p* < 0.05) following Duncan’s post hoc means-comparison test.

**Figure 6 foods-10-01761-f006:**
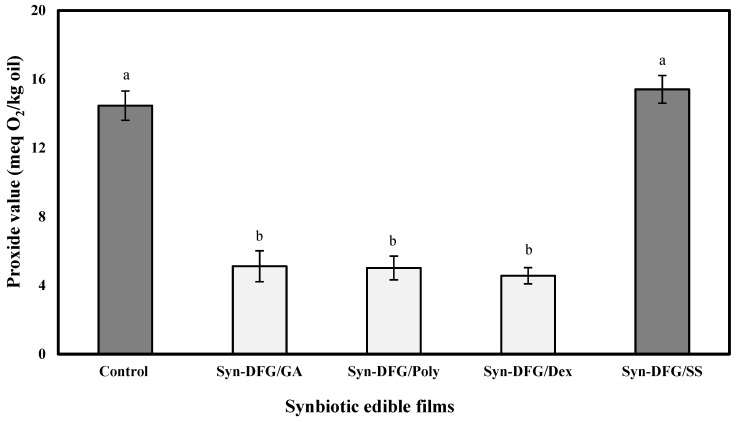
Prevention against oxidation of synbiotic edible films. a–b: Different letters on each column indicate significantly different values (*p* < 0.05) following Duncan’s post hoc means-comparison test.

**Table 1 foods-10-01761-t001:** Inactivation rates of *L. casei* immobilized in plasticized gelatin matrices (probiotic films) stored under room temperature condition.

Probiotic Film	k 25 °C (%/day^−1^)	Estimated Shelf-Life at 25 °C (day)	R^2^
Probiotic-DFG	4.47 ± 0.12 ^a^	20.11	0.957
Probiotic-BG	4.66 ± 0.12 ^a^	19.27	0.908
Probiotic-FG	4.72 ± 0.12 ^a^	19.03	0.887

^a^ The same letter indicates no significantly different values (*p* > 0.05) according to Duncan’s post hoc means-comparison test. Data are presented as mean ± SD (*n* = 3). Shelf-life mentions to the time needed to induce the fate of the 90% of total viable cells of *L. casei*. DFG: Duck feet gelatin, FG: Fish gelatin, BG: Bovine gelatin.

**Table 2 foods-10-01761-t002:** Optical and color properties of edible synbiotic films containing *L. casei* and different types of prebiotic fibers.

Synbiotic Film	L^*^	a^*^	b^*^	ΔE^*^	*Opacity Values*
Control	95.35 ± 0.07 ^a^	−0.39 ± 0.01 ^d^	3.42 ± 0.01 ^e^	-	0.82 ± 0.04 ^c^
Syn-DFG/Dex	94.17 ± 0.08 ^c^	−0.05 ± 0.003 ^a^	4.41 ± 0.01 ^c^	3.9 ± 0.02 ^c^	1.79 ± 0.15 ^b^
Syn-DFG/GA	94.59 ± 0.08 ^b^	−0.33 ± 0.02 ^c^	4.78 ± 0.05 ^b^	4.33 ± 0.06 ^b^	3.94 ± 0.11 ^a^
Syn-DFG/Poly	94.51 ± 0.1 ^b^	−0.74 ± 0.03 ^e^	6.24 ± 0.08 ^a^	5.74 ± 0.1 ^a^	3.24 ± 0.08 ^a^
Syn-DFG/SS	94.47 ± 0.12 ^b^	−0.25 ± 0.02 ^b^	3.80 ± 0.04 ^d^	3.32 ± 0.07 ^d^	1.92 ± 0.12 ^b^

^a–e^ Different letters between rows indicate significantly different values (*p* < 0.05) according to Duncan’s post hoc means-comparison test. Data are presented as mean ± SD (*n* = 3). Duck feet gelatin, Dex: Dextrin, GA: gum Arabic, Poly: polydextrose, SS: sago starch.

**Table 3 foods-10-01761-t003:** Inactivation rates of *L. casei* immobilized in plasticized synbiotic DFG and stored either in chilled or room-temperature conditions.

Synbiotic Film	k 4 °C (%/day^−1^)	Estimated Shelf- Life * at 4 °C	R^2^	k 25 °C (%/day^−1^)	Estimated Shelf- Life at 25 °C	R^2^
Control	1.14 ± 0.09 ^c^	78	0.95	4.49 ± 0.12 ^b^	20	0.95
Syn-DFG/Dex	0.9 ± 0.05 ^b^	99	0.96	3.33 ± 0.08 ^a^	27	0.97
Syn-DFG/Poly	0.9 ± 0.03 ^b^	99	0.97	4.28 ± 0.05 ^b^	21	0.97
Syn-DFG/GA	0.8 ± 0.04 ^a^	111	0.97	3.1 ± 0.07 ^a^	29	0.99
Syn-DFG/SS	0.95 ± 0.08 ^b^	94	0.95	4.31 ± 0.11 ^b^	21	0.97

^a–c^ Different letter between rows indicate significantly different values (*p* < 0.05) according to Duncan’s post hoc means-comparison test. Data are presented as mean ± SD (*n* = 3). *: Shelf-life mentions to the time needed to induce the fate of the 90% of total viable cells of *L. casei*.

**Table 4 foods-10-01761-t004:** Physicochemical properties of synbiotic edible films.

Synbiotic Film	Thickness (mm)	MC (%)	SR (%)	WS (%)
Control	0.17 ± 0.01 ^c^	10.15 ± 0.1 ^b^	662.1 ± 13.2 ^a^	19.37 ± 1.12 ^c^
Syn-DFG/Dex	0.21 ± 0.05 ^a^	9.88 ± 0.21 ^b^	444.34 ± 10.1 ^c^	18.08 ± 0.95 ^c^
Syn-DFG/Poly	0.21 ± 0.02 ^a^	10.64 ± 0.15 ^b^	675.55 ± 19.8 ^a^	44.07 ± 1.4 ^a^
Syn-DFG/GA	0.19 ± 0.01 ^b^	10.19 ± 0.11 ^b^	602.34 ± 9.2 ^b^	29.49 ± 0.78 ^b^
Syn-DFG/SS	0.21 ± 0.01 ^a^	10.95 ± 0.05 ^a^	569.32 ± 12.11 ^b^	18.57 ± 1.1 ^c^

^a–c^ Different letter between rows indicate significantly different values (*p* < 0.05) following Duncan’s post hoc means-comparison test. Data are shown as mean ± SD (*n* = 3). MC: Moisture content, SR: Percentage of water gained [g]/total solids [g], WS: percentage of soluble solids [g]/total solids [g].

**Table 5 foods-10-01761-t005:** Mechanical properties including of tensile strength (TS), elongation at break (EB), and Young’s modulus (YM) of synbiotic DFG films.

Synbiotic Film	TS (MPa)	EB (%)	YM (MPa)
Control	17.1 ± 2.2 ^ab^	31.3 ± 1.2 ^c^	516.81 ± 11.2 ^ab^
Syn-DFG/Dex	18.36 ± 0.18 ^a^	27.93 ± 3.2 ^c^	574.66 ± 12.2 ^a^
Syn-DFG/Poly	6.97 ± 0.5 ^c^	94.64 ± 6.4 ^a^	78.89 ± 8.7 ^c^
Syn-DFG/GA	18.2 ± 0.22 ^a^	49.53 ± 4.1 ^b^	407.2 ± 10.2 ^b^
Syn-DFG/SS	15.01 ± 0.9 ^b^	48.52 ± 2.5 ^b^	413.04 ± 19.41 ^b^

^a–c^ Different letters indicate significantly different values (*p* < 0.05) following Duncan’s post hoc means-comparison test. Data are shown as mean ± SD (*n* = 3).

## Data Availability

The data presented in this study are available on request from the corresponding author.

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
