# Peer review of "Characterization and Cell Viability of Probiotic/Prebiotics Film Based on Duck Feet Gelatin: A Novel Poultry Gelatin as a Suitable Matrix for Probiotics"

_foods, 2021, doi:10.3390/foods10081761_

Round 1

Reviewer 1 Report

Dear Authors,

The manuscript is interesting, however it is important to provide more data about practical application of your idea.

1. Line 53: this is concentration. Please also provide data concerning the daily dose.
2. Line 65, 66: 10^8-10^9, 10^6-10^7, colon content.
3. The plate culture is a gold standard. What do you think about the application of Flow Cytometry especially to obtain faster results? 
4. What is the practical value of the study? The shelf life of probiotics in the finnished product should be min. 18 months in temperature 15-25 C. What shelf-life could be expected in case of DFG?
5. Please compare the economic aspects of DFG in comparison with other methods.

Author Response

Dear Reviewer, Thanks for all your comments, please find our reply in the attached file.

Reviewer 2 Report

The work needs an overall revising of the English used, as it is very confusing and hard to read

The abstract is confusing and hard to understand. It would benefit from a revision.

The introduction is badly structured which makes for a difficult read. Please revise.

You used DFG but you make no references to this compound/product in the introduction. What’s its relevance?

What are the characteristics of the dextrin and polydextrose obtained from Sigma?

DFG was extracted from where?

Why do you use fish gelatin?

What are the characteristics of Gum Arabic and sago starch?

What is low oxygen tension?

Section 2.2 is very confusing. It must be re-written.

Quality of the figures presented is low

Author Response

(The authors gave the same response as above.)

Reviewer 3 Report

The present manuscript is dealing with the prebiotic and probiotic films based on duck feet gelatin (DFG). The main issue concerns the presentation of the results, both for the quality of the English language and for the clarity of the presentation. Results understanding is very hards. The results description and discussion must be re-arranged (I suggest keeping separate Results Section and Discussion Section). Many acronyms are used, some of which are not explained within the manuscript.

Furthermore, another critical point concerns the improving effect of the viability of the DFG compared to FG and BG (Fig. 1). From this assumption, the study follows a new direction by testing only the DFG with Dex, GA, Poly, SS. But is the DFG effect really better than FG and BG? Are there any statistical differences? Looking at Fig. 1, the protective effect does not seem particularly improved.

Furthermore, the preservative effect of DFG appears very scarce. In the past, microencapsulation and storage solutions at room temperatures have been proposed with better promising results [Heidebach, T., Först, P., & Kulozik, U. (2010). Influence of casein-based microencapsulation on freeze-drying and storage of probiotic cells. Journal of food engineering, 98(3), 309-316].

  • Abstract: “Four synbiotic systems were used to immobilization probiotic (Lactobacillus casei ATCC), including prebiotics (dex-28 trin, polydextrose, gum Arabic, and sago starch), DFG and glycerol”: Please rephrase, I think the sentences can be misunderstood. Prebiotics, DFG and glycerol are three of the tested systems?
  • P2 L65-66: please, report correctly the exponent of the power
  • P2 L91-93: please, report correctly the exponent of the power
  • P3 L102: please, Italicize casei.
  • P3 L131: please, explain SEFs.
  • P3 L137: please, explain FFSs.
  • P4 L160: please, Italicize L. casei.
  • P5 L235: please describe the control sample
  • P7 L292-293: “As the curve shows, DFG has more 292 potential to keep this probiotic alive”: are the differences statistically significant?
  • P7 L300: “…during the matrix” what? Please, rephrase.
  • P7 L300: “Soukoulis, et al. [39] was evaluated composition…” Please, rephrase.
  • P7 L302-303: “were revealed that gelatin..” Please, rephrase.
  • P7 L314-315: “Measuring appearance properties by measuring color provides information about the films intended application, not the chemical reaction”. Please, rephrase.
  • P7 L318: “Table 1 is displayed” Please, rephrase.
  • P7 L318-322: “Table 1 is displayed the optical and colour properties of edible bioactive films. It shows that the addition of prebiotic fiber was associated with a detectable decrease (p < 0.05) of the transparency (or increase in opacity) of the edible films compared to the only DFG containing ones”: I don't see any transparency (opacity) data in Table 1 or within the manuscript.
  • P10 L384-385: “The GA was the highest estimated shelf-life values” Please, rephrase.

Author Response

(The authors gave the same response as above.)

Round 2

Reviewer 2 Report

In line 124 you say that L.casei was grown in the absence of oxygen. Why? DSMZ datasheet for this microorganism says 95% air with 5% CO2.

In line 157 you write “Six pellets of L. casei (corresponding to 300 mL of culture MRS broth) were added to each FFSs (100 mL)”. How can this be? You centrifuged the culture. How can you from a cellular pellet say it corresponds to 300 mL of culture? Furthermore, what is this supposed to mean? The correct way would be to standardize in inoculum mass.

In line 164 how did you evaluate the brittleness and flexibility?

Why in the microbiological analysis section the different films were processed in different ways? Film processing has to be standardized for results to be comparable.

In table 1 what are the units of the estimated shelf-life?

Where is the discussion of the appearance properties?

Author Response

Respond to Reviewer’s Comments in purple colour in the text:

Comment 1:

In line 124 you say that L.casei was grown in the absence of oxygen. Why? DSMZ datasheet for this microorganism says 95% air with 5% CO2.

Respond to Comment 1:

Dear reviewer, many thanks for your great attention. It was corrected as the reference and deleted the phrase of “the absence of oxygen”. Please refer to lines 125-125.

Comment 2: In line 157 you write “Six pellets of L. casei (corresponding to 300 mL of culture MRS broth) were added to each FFSs (100 mL)”. How can this be? You centrifuged the culture. How can you from a cellular pellet say it corresponds to 300 mL of culture? Furthermore, what is this supposed to mean? The correct way would be to standardize in inoculum mass.

Respond to Comment 2:

Many thanks for your comment. We revised this part and was corrected the error following other references and added the correct explanation of the accurate value to secure the number of inoculated probiotic in the 100 mL film solution which is in lines 156-157, as you suggested.

Comment 3: In line 164 how did you evaluate the brittleness and flexibility?

Respond to Comment 3: Dear reviewer, many thanks for this precious comment; since in the next steps all the mechanical tests of the synbiotic films have been done, it was not necessary to mention this sentence in this section and we deleted it, as your comment.

Comment 4: Why in the microbiological analysis section the different films were processed in different ways? Film processing has to be standardized for results to be comparable.

Respond to Comment 4:

Many thanks for this great question. To clarify the issue, we must say that we have had two separate study phases, the results of which have been compared separately. First, in the probiotic film (probiotic inoculated in the gelatins based), a kind of comparison of the survival of the bacteria tested in the base matrix (DFG, fish and bovine gelatins) has been done, and after confirming that: "this matrix tested (DFG) has the ability to survive probiotic", we have made synbotic films, so, these methods are different for probiotic and symbiotic films and the results of the same methods have only been compared.

Comment 5: In table 1 what are the units of the estimated shelf-life?

Respond to Comment 5:

Dear reviewer, Thanks for this comment. We added the unit of (day) in the table 1, as you suggested.

Comment 6: Where is the discussion of the appearance properties?

Respond to Comment 6:

Dear reviewer, Thanks for this comment. Figure 2, which shows the appearance of the films, actually confirms the measurement results of the color components. In fact, this Fig is provided to confirm lines 361-380 about the color characteristics of synbiotics.

Thank you so much giving us another chance to revise
